



# Micro-spectroscopic and freezing characterization of ice-nucleating particles collected in the marine boundary layer in the Eastern North Atlantic

Daniel A. Knopf[1], Joseph C. Charnawskas[1], Peiwen Wang[1], Benny Wong[1], Jay M. Tomlin[2], Kevin A. Jankowski[2], Matthew Fraund[3], Daniel P. Veghte[4†], Swarup China[4], Alexander Laskin[2], Ryan C. Moffet[5], Mary K. Gilles[6], Josephine Y. Aller[1], Matthew A. Marcus[7], Shira Raveh-Rubin[8], Jian Wang[9]

[1]School of Marine and Atmospheric Sciences, Stony Brook University, Stony Brook, NY 11794, USA
[2]Department of Chemistry, Purdue University, West Lafayette, IN 47907, USA
[3]Ultrafast X-Ray Science Laboratory, Lawrence Berkeley National Laboratory, Berkeley, CA 94720, USA
[4]Environmental Molecular Sciences Laboratory/Pacific Northwest National Laboratory, Richland, WA 99354, USA
[5]Sonoma Technology, Inc., Petaluma, CA 94954, USA
[6]Chemical Sciences Division, Lawrence Berkeley National Laboratory, Berkeley, CA 94720, USA
[7]Advanced Light Source, Lawrence Berkeley National Laboratory, Berkeley, CA 94720, USA
[8]Department of Earth and Planetary Sciences, Weizmann Institute of Science, Rehovot 76100, Israel
[9]Center for Aerosol Science and Engineering, Department of Energy, Environmental and Chemical Engineering, Washington University in St. Louis, St. Louis, MO 63130, USA

†Current address: Center for Electron Microscopy and Analysis, The Ohio State University, Columbus, OH 43212, USA

*Correspondence to*: Daniel A. Knopf (daniel.knopf@stonybrook.edu)

**Abstract.** Formation of atmospheric ice plays a crucial role in the microphysical evolution of mixed-phase and cirrus clouds and thus climate. How aerosol particles impact ice crystal formation by acting as ice-nucleating particles (INPs) is a subject of intense research activities. To improve understanding of atmospheric INPs, we examined daytime and nighttime particles collected during the Aerosol and Cloud Experiments in the Eastern North Atlantic (ACE-ENA) field campaign conducted in summer 2017. Collected particles, representative of a remote marine environment, were investigated for their propensity to serve as INPs in the immersion freezing (IMF) and deposition ice nucleation (DIN) modes. The particle population was characterized by chemical imaging techniques such as computer-controlled scanning electron microscopy with energy dispersive X-ray analysis (CCSEM/EDX) and scanning transmission X-ray microscopy with near-edge X-ray absorption fine structure spectroscopy (STXM/NEXAFS). Four major particle-type classes were identified where internally mixed inorganic-organic particles make up the majority of the analyzed particles. Following ice nucleation experiments, individual INPs were identified and characterized by SEM/EDX. The identified INP types belong to the major particle-type classes consisting of fresh sea salt with organics or processed sea salt containing dust and sulfur with organics. Ice nucleation experiments show IMF events at temperatures as low as 231 K including the subsaturated regime. DIN events were observed at lower temperatures of 210 to 231 K. IMF and DIN observations were analyzed with regard to activated INP fraction, ice-nucleation active sites (INAS) densities, and water activity-based immersion freezing model (ABIFM) yielding heterogeneous ice nucleation rate coefficients. Observed IMF and DIN events of ice formation and corresponding derived freezing rates



demonstrate the marine boundary layer aerosol particles can serve as INPs under typical mixed-phase and cirrus clouds conditions. The derived IMF and DIN parameterizations allow for implementation in cloud and climate models to evaluate predictive effects of atmospheric ice crystal formation.

**Short Summary (500 character in total)**

Marine boundary layer aerosols collected in the remote region of the eastern north Atlantic induce immersion freezing and deposition ice nucleation under typical mixed-phase and cirrus clouds conditions. Corresponding ice nucleation parameterizations for model applications have been derived. Chemical imaging of ambient aerosol and ice-nucleating particles demonstrates that the latter is dominated by sea salt and organics, while also representing a major particle type in the particle
population.

**Introduction**

Understanding how atmospheric aerosol serves as INPs is necessary to advance our understanding of cloud microphysical processes that impact precipitation and climate (Boucher et al., 2013;Storelvmo, 2017;Baker and Peter, 2008;Mülmenstädt et al., 2015;Lohmann and Feichter, 2005). To improve our predictive capabilities of atmospheric ice formation, laboratory and
field-based research addressing the efficacy and physicochemical nature of INPs has intensified over recent years (Cantrell and Heymsfield, 2005;Murray et al., 2012;Kanji et al., 2017;Cziczo et al., 2017;DeMott et al., 2011;Knopf et al., 2018;Knopf et al., 2021). The challenge lies in the fact that only a small fraction of total aerosol particles serve as INPs (DeMott et al., 2010). Furthermore, different ice formation pathways exist, which in turn depend on ambient conditions such as temperature (*T*) and relative humidity (RH) (Pruppacher and Klett, 1997;Vali et al., 2015;Knopf et al., 2018). To contribute to our current
understanding of atmospheric INPs, we performed aerosol particle collection, compositional analysis, and subsequent ice nucleation experiments as part of the Aerosol and Cloud Experiments in the Eastern North Atlantic (ACE-ENA) field campaign (Wang et al., 2021).

The motivation of the ACE-ENA campaign was to further our understanding of marine boundary layer clouds through field measurements of cloud condensation nuclei (CCN), drizzle, and cloud microphysics (Wang et al., 2021). This campaign
involved ground site measurements at the ENA site and airborne instrument employment during two intensive operational periods in summer 2017 and winter 2018. The ground ENA site was established on Graciosa Island in the Azores, Portugal (39° 5' 30" N, 28° 1' 32" W, 30.48 m above mean sea level, Fig. S1) by the U.S. Department of Energy (DOE) Atmospheric Radiation Measurement (ARM) Climate Research Facility (Wang et al., 2021;Mather and Voyles, 2013). This site experiences a wide range of meteorological and cloud conditions and can be prone to aerosol particles downwind of the North American
continent (Wang et al., 2021). Furthermore, dry intrusion events from the free troposphere into the marine boundary layer can cause its deepening, drying, and cooling (Raveh-Rubin, 2017;Ilotoviz et al., 2021) and impact the make-up of the boundary



layer aerosol population (Tomlin et al., 2021). In this study, we report on composition and ice nucleation analyses of particles collected at the ENA ground site during the first intensive operating period during the summer of 2017.

Laboratory and field studies have demonstrated large variety of particle types that can act as INPs. Inorganic particles such as mineral dust, metal oxides, and crystalline salts can serve as INPs (Murray et al., 2012;Kanji et al., 2017;Cziczo et al., 2017). Organic particles including primary organic aerosol (POA), e.g., from marine environments, biomass burning, and fossil fuel combustion, and secondary OA (SOA) from biogenic and anthropogenic precursor gases have been shown to initiate ice formation (Kanji et al., 2017;Knopf et al., 2018). Biological particles such as bacteria, fungal, spores, phytoplankton, and soil dust particles from agricultural lands can contribute to INPs (Murray et al., 2012;Kanji et al., 2017;Knopf et al., 2018;Despres et al., 2012). The majority of studies have focused on immersion freezing (IMF), where the immersed INPs initiate freezing of the supercooled droplets, compared to deposition ice nucleation (DIN), where ice forms from the supersaturated gas phase on the surface of the INPs (Knopf et al., 2018;Vali et al., 2015;Pruppacher and Klett, 1997). DIN may also be the consequence of homogeneous freezing of water in nanometer-sized pores due to a depression of the adjacent saturation vapor pressure, termed pore condensation freezing (PCF) (David et al., 2019;Marcolli, 2014, 2020).

The ENA ground site observatory is located in a remote marine region and close to the island shore (Fig. S1). For this reason, the boundary layer particle types likely constitute a large fraction of sea spray aerosol (SSA) particles. Aged and nascent SSA particles consist of inorganic and organic species, where smaller particle sizes often display greater enrichment of organic species (Cochran et al., 2017;Pham et al., 2017;Jayarathne et al., 2016;Aller et al., 2017;Facchini and O'Dowd, 2009;O'Dowd et al., 2004;Laskin et al., 2016;Ault et al., 2013). Field and laboratory experiments have demonstrated that SSA particles can act as INPs where aerosolized organic carbonaceous (OC) matter associated with exudates from phytoplankton serves as ice-nucleating agents (Creamean et al., 2019;McCluskey et al., 2018a;McCluskey et al., 2018b;McCluskey et al., 2017;DeMott et al., 2016;Knopf et al., 2011;Alpert et al., 2011b;Alpert et al., 2011a;Ladino et al., 2016;Wilson et al., 2015;Prather et al., 2013a;Schnell, 1975;Irish et al., 2019;Irish et al., 2017;Knopf et al., 2014;Ickes et al., 2020;Wilbourn et al., 2020;Wolf et al., 2019;Schnell and Vali, 1975;Roy et al., 2021;Wagner et al., 2021).

Micro-spectroscopic single-particle analyses of the collected particle population and of INPs allow for a better understanding of how a particle population serves as a source of INPs and which particles preferentially act as INPs (China et al., 2017;Knopf et al., 2014;Wang et al., 2012b;Hiranuma et al., 2013;Cziczo et al., 2017;Knopf et al., 2018). Nanoscale chemical imaging methods such as computer-controlled scanning electron microscopy with energy dispersive X-ray analysis (CCSEM/EDX) and scanning transmission X-ray microscopy with near-edge X-ray absorption fine structure spectroscopy (STXM/NEXAFS) are typically employed to examine the elemental composition of a large number of aerosol particles and to determine the organic speciation and mixing state of collected particles, respectively. Examination of large ensembles of individual particles by CCSEM/EDX allows for a statistically significant representation of the major particle-type classes present in a particle population (Knopf et al., 2014;Wang et al., 2012b;Thompson, 1987;Laskin and Cowin, 2001;Hopkins et al., 2007a;Hopkins et al., 2007b;Laskin et al., 2019;Laskin et al., 2016;Laskin et al., 2006). STXM/NEXAFS is applied to infer the particle population mixing state (MS) and the individual particle's composition and organic volume fraction (OVF), and





to examine the nature of the organic carbon associated with the particle (Laskin et al., 2019;Moffet et al., 2016;Laskin et al., 2016;O'Brien et al., 2015;Moffet et al., 2010b;Tivanski et al., 2007;Bluhm et al., 2006;Knopf et al., 2021).

This study follows our previous analytical methods of ambient aerosol and INPs (Knopf et al., 2018;China et al., 2017;Knopf et al., 2014;Wang et al., 2012b;Knopf et al., 2010;Knopf et al., 2021). Here, we collected particle samples over several days and nights at the ENA site during summer 2017 for analysis of ambient aerosol and INPs. We apply CCSEM/EDX and STXM/NEXAFS for identification and characterizations of the particle population present on the samples. We measure IMF and DIN for typical tropospheric RH and $T$ as low as 210 K. We identify individual INPs using SEM/EDX and discuss those findings in context of particle population present on the samples. We derive IMF and DIN kinetics, report INP $L^{-1}$ of air, ice nucleation active sites (INAS), and classical nucleation theory (CNT) based parameters for each ice formation pathway and provide corresponding IMF and DIN parameterizations for application in cloud and climate models.

## 2. Experimental methods

### 2.1 Particle sampling

Particles were collected by impaction using a Multi Orifice Uniform Deposition Impactor (MOUDI, 110-R). Since ambient particle numbers were low and sampling intervals for this ice nucleation study were limited, particle collection was conducted over several days intermittently. This ensured sufficient aerosol loading on substrates for single-particle chemical analyses and ice nucleation experiments based on our previous studies (China et al., 2017;Knopf et al., 2014;Wang et al., 2012b;Wang et al., 2012a). Hence, particle analysis and INP characterization should not be interpreted as single defined events but as an average over 2-4 days. Table 1 lists the particle collection characteristics of the four samples (two daytime and two nighttime samples). The particle samples were collected on the 6th stage of the MOUDI with $D_{50\%} = 0.56$ µm. Pre-arranged substrates were mounted on the MOUDI stage: $Si_3N_4$ coated silicon wafer chips for ice nucleation experiments, transmission electron microscopy (TEM) grids (copper 400 mesh grids, carbon type B film, Ted Pella, Inc.) for CCSEM/EDX and STXM/NEXAFS analyses (Knopf et al., 2014;Charnawskas et al., 2017). The particle samples were stored in airtight sealed container at room temperature under dry conditions (Knopf et al., 2014;Tomlin et al., 2021).

Figure 1 depicts exemplary backward trajectory calculations for the collection time periods given in Table 1. (Supplementary Fig. S2 provides more backward trajectory calculations for the 2-4 days sampling periods.) The trajectories were calculated using global wind data from the ECMWF reanalyses ERA5 (Hersbach et al., 2020). The data are horizontally interpolated to a 0.5° × 0.5° grid, with 3-hourly time intervals and 137 vertical hybrid levels. The Lagrangian analysis tool LAGRANTO) version 2.0 (Sprenger and Wernli, 2015) was applied to calculate the trajectories from the measurement site, backwards for 10 days. Trajectories were calculated from the site at all available model levels from ground to a pressure level 50 hPa below ground pressure, resulting in 12 backward trajectories for each calculation (sampling time step). The black dots in Fig. 1 represent the location of the air masses 5 days prior to their arrival at the site. In addition, $T$ and RH were traced along the trajectory positions (Fig. 1). These additional parameters indicate that the air parcel, a few days before arrival at





measurement site, experienced RH > 50%. It is worth noting that no dry intrusions (dry, deeply descending airstreams from the upper troposphere toward the marine boundary layer) according to the Lagrangian descent criterion in Raveh-Rubin (2017)

were detected during the collection time periods (Tomlin et al., 2021;Ilotoviz et al., 2021).

## 2.2 Micro-spectroscopic single-particle analysis

We employed chemical imaging methods to analyse ensembles of particles collected on substrates and INPs following our previous work (China et al., 2017;Knopf et al., 2014;Wang et al., 2012b;Moffet et al., 2013;Moffet et al., 2010b;Knopf et al., 2010;Hopkins et al., 2008;Hopkins et al., 2007a;Laskin et al., 2019;Laskin et al., 2016;Laskin et al., 2002;O'Brien et al.,

2015;Tomlin et al., 2021), thus our approach is only described briefly here. Characterization of the particle ensembles was performed by probing particle composition by CCSEM/EDX and STXM/NEXAFS. SEM/EDX was applied to infer the elemental composition of identified INPs. STXM/NEXAFS was used to give molecular information of particle carbonaceous content. For CCSEM/EDX and STXM/NEXAFS, we examined particles with an equivalent circle diameter larger than 200 nm.

CCSEM/EDX was operated at 20 kV (CCSEM/EDX; FEI Quanta 3D, EDAX Genesis) to determine the size-resolved particle-type distribution in the ambient aerosol population using machine learning $k$-means cluster analysis (e.g., Knopf et al., 2014;Tomlin et al., 2021;Laskin et al., 2005;Moffet et al., 2012;Tomlin et al., 2020). Analysis of EDX particle spectra allowed to quantify relative atomic fractions of the following elements C, N, O, Na, Mg, Al, Si, P, S, Cl, K, Ca, Mn, and Fe. This type of analysis allows to identify major particle-type classes that are significantly different in their elemental composition. Table

2 provides the number of particles and mean equivalent circle diameter for each particle sample. More than 2000 particles were examined for each sample, though for identification of the major particle-type classes via $k$-means cluster analysis, the CCSEM/EDX data for all samples were taken together. Such a large sample ensures a significant representation of the actual ambient particle population (Wang et al., 2012b;Thompson, 1987). For each particle sample, the distribution of the different particle-types for examined equivalent circle diameters were derived. CCSEM/EDX allows for determination of the particle

number density present on the substrates and the average particle diameter. Particle size information provides an estimate of the total particle number and surface area present during an ice nucleation experiment.

SEM/EDX was employed to characterize individual INPs spotted in optical microscopy (OM) ice nucleation experiments similar to our previous work (China et al., 2017;Knopf et al., 2014;Lata et al., 2021). Ice crystal formation by INPs was first recorded by OM at different magnifications as shown in Fig. S3 and as outlined below. Then, digital pattern recognition and

triangulation allowed to relocate the INP in the SEM for imaging and elemental composition analysis (Knopf et al., 2014;China et al., 2017).

We applied STXM/NEXAFS (Kilcoyne et al., 2003) to examine the particle carbon-specific MS, organic carbon (OC) speciation, and OVF. STXM/NEXAFS was performed at the carbon K-edge (278-320 eV) to allow for chemical characterization of the particle composition and speciation of the OM fraction following our previous studies (Knopf et al.,

2014;Moffet et al., 2010b;Moffet et al., 2010a;Hopkins et al., 2007b;Tomlin et al., 2020;Fraund et al., 2020;Laskin et al.,



2019;Moffet et al., 2016;O'Brien et al., 2014;Moffet et al., 2013;Tomlin et al., 2021). In brief, two types of measurements were conducted. To examine a larger number of particles and their associated morphology and mixing state, STXM images at 35 nm pixel and 1 ms dwell time were recorded at four selected energies: 278 eV (pre-edge, inorganic: IN), 285.4 eV (C=C, elemental carbon EC), 288.5 eV (COOH, organic carbon OC), and 320 eV (post-edge). The recorded intensity across the

particle segments for each energy were evaluated as optical density (OD). IN rich regions were identified by having $OD_{278eV}$ / $OD_{320eV} > 0.5$. Total organic carbon was derived from the difference between $OD_{320eV}$ and $OD_{278eV}$. Presence of carboxylic acid (COOH) was evaluated by the difference between $OD_{288.5eV}$ and $OD_{278eV}$. Enrichment of elemental carbon (EC) was assessed following method described in  (Hopkins et al., 2007b). These measurements allow for chemical composition mapping and derivation of the OVF of the individual particles (Fraund et al., 2019). For the OVF calculations, we used NaCl

and adipic acid serving as representative mass absorption coefficients for the IN and OC particle fractions, respectively. These data were then applied to derive size-resolved mixing state and OVF distributions of the particle populations. The second type of STXM measurement involves the acquisition of high-resolution energy absorption (NEXAFS) spectra at 35 nm pixel resolution and 1 ms dwell time, where the particles were examined at 96 different energies between 278 and 320 eV. These NEXAFS spectra allow for a more detailed interpretation of the nature of the particulate OC.

**2.3 Ice nucleation experiment**

We employ a custom built vapor-controlled cryo-cooling stage consisting of an ice nucleation cell and an OM to determine under which temperature and RH ice formation commences in the IMF and DIN modes as previously described (China et al., 2017;Charnawskas et al., 2017;Knopf et al., 2014;Wang et al., 2012a;Knopf et al., 2011;Alpert et al., 2011b;Wang and Knopf, 2011). The particle samples were mounted into the ice nucleation cell that allows control of particle temperature from 200 K

to room temperature and to exposure to humidity up to saturation. A humidified $N_2(g)$ ultra-high purity (UHP) flow of about 1 standard liter per minute (SLPM) is supplied into the ice nucleation cell. A chilled mirror hygrometer continuously measures the dew point temperature at the exit of the cell. RH in the ice nucleation cell is derived by the particle temperature and the measured dewpoint (Wang and Knopf, 2011). Visual observation of the ice formation event allows to distinguish between IMF and DIN (Wang and Knopf, 2011). Calibration of the humidity is performed by controlled ice crystal growth and sublimation

experiments. The temperature accuracy of the cooling stage is independently verified by measuring the melting points of different organic compounds and ice (Knopf and Rigg, 2011). Typical uncertainty in $RH_{ice}$ is about ±4 to ±5.7% and ±0.15 K for temperature. For each sample and investigated temperature at least three experiments were conducted. A typical experiment starts at a given particle temperature and for subsaturated conditions with respect to ice (i.e., $RH_{ice} < 100\%$) of about 30-60%. Subsequently, $RH_{ice}$ is continuously increased by about 1.5-2.3% per minute, reflecting actual vertical updraft velocities

experienced in cirrus cloud formation, by decreasing the particle temperature by 0.1 K min[-1] (Wang and Knopf, 2011;Knopf and Koop, 2006). Every 0.02 K or 12 s an image of the experiment and associated particle temperature and dew point are recorded. This allows for the analysis of ice nucleation mode, freezing temperature and humidity, INP activated fraction, INAS





density, and ice nucleation rate coefficient (China et al., 2017;Alpert et al., 2011b). It furthers enables detection of the ice nucleation mode (IMF and DIN) and identification of the individual INP (Fig. S3).

**2.4 Ice nucleation analysis**

The efficacy of INPs is typically expressed by different freezing parameterizations (Knopf et al., 2021;Knopf et al., 2018;Knopf et al., 2020;Vali, 1971;Connolly et al., 2009;DeMott et al., 2010;Knopf and Alpert, 2013). This includes the number concentration of INPs activated per liter of air for a given $T$ and RH. Another commonly employed parameterization is the INAS density, $n_s$, in units cm$^{-2}$ (Connolly et al., 2009). Both of these types of ice nucleation parameterizations are based

on the deterministic or singular hypothesis approach (Vali, 1971;Connolly et al., 2009). This approach assumes that $n_s$ depends only on $T$, thereby neglecting time, and that each nucleation event is associated with a characteristic ice nucleation active site. Typically, $n_s(T)$ is reported for saturated conditions, i.e., an INP is engulfed in water (Murray et al., 2012). In contrast, CNT assumes that nucleation is time dependent and, thus, is stochastic in nature (Pruppacher and Klett, 1997;Knopf et al., 2020). Its time dependence yields an ice nucleation rate derived from the heterogeneous ice nucleation rate coefficient $J_{het}$ (cm$^{-2}$ s$^{-1}$).

The water activity ($a_w$) based immersion freezing model (ABIFM) expresses $J_{het}$ as a function of the water activity criterion $\Delta a_w$, thereby accounting for $T$ and RH conditions including the subsaturated regime (Knopf et al., 2020;Alpert and Knopf, 2016;Knopf and Alpert, 2013). $\Delta a_w$ represents the difference between the water activity at the ice melting point, $a_w^{ice}$, for the observed freezing temperature and the water activity at the temperature for which ice formation was observed: $\Delta a_w(T) = a_w^{ice}(T) - a_w(T)$ (Koop et al., 2000;Knopf and Alpert, 2013). Assuming that the particle is in equilibrium with the gas phase,

particle $a_w$ = RH. Hence, $\Delta a_w(T)$ depends on two variables, $T$ and RH. Application of either INAS or ABIFM to experimental freezing data acquired at given conditions (i.e., $T$, RH, nucleation time) yield the same INP number concentrations. However, when applied to conditions different from experiments, e.g., when looking at different cloud activation time scales or particle size distributions, differences in predicted INP number concentrations by these parameterizations are expected, though the magnitude of difference depends on the activation rate and the efficacy of the INPs (Knopf et al., 2021).

**3. Results and discussion**

**3.1 Particle population characterization**

Figure 2 show the *k*-means cluster analysis results derived from CCSEM/EDX measurements. Over 9000 individual particles from two daytime and two nighttime samples were analyzed. This analysis yielded 4 significantly different particle types. According to the cluster-averaged elemental composition given by the atomic percentage we have termed those particle

types as: i) processed sea salt with signatures of dust, sulfur, and OC (sea salt proc./dust/org), ii) fresh sea salt (sea salt), iii) processed sea salt with dust signatures (sea salt proc./dust), and iv) organic with chlorine (organic/Cl). Processed sea salt indicates the loss of chlorine, likely due to chemical reactions with particulate nitric acid, sulfuric acid, and organic acids





leading to gaseous HCl (Wang et al., 2015;Laskin et al., 2012a;Angle et al., 2021). Particle type 'sea salt proc./dust/org' contains the most sulfur of all particle types and shows a greater amount of carbon and oxygen compared to particle type 'sea
salt proc./dust'. Particle type 'sea salt' indicates fresh sea salt particles with a Na/Cl ratio close to unity. Lastly, particle type 'organic/Cl' represents particles with a greater atomic fraction of OC and chlorine. The naming of the different particle-type classes is arbitrary. It serves mostly the purpose to assess if identified INPs belong to those 4 major particle-type classes or if they belong to completely different particle types as will discussed below. Table 2 provides additional information on the number of particles per sample investigated and the average area equivalent diameter (AED) for each sample. The mean AED
of all examined particles were in the submicrometer range. Overall, the most abundant particle type was 'fresh sea salt' particles, followed by 'sea salt proc./dust', 'sea salt proc./dust/org', and 'organic/Cl'.

Figure 3 displays the CCSEM/EDX derived particle-type size distribution derived from MOUDI stage 6 for the four particle samples. Day 1 and 2 and Night 1 samples show the dominance of fresh sea salt particles and differ only in the amounts of 'sea salt proc./dust' and 'sea salt proc./dust/org' particles. Day 2 sample shows the greatest amount of 'sea salt
proc./dust/org' particles. For all samples, for particles larger than 1 µm, few 'sea salt proc./dust' particles are present. Night 2 sample displays a very different particle type distribution. Fresh 'sea salt' is not present, and the examined particle size range is dominated by 'sea salt proc./dust/org' particles. It is also the only sample that contains significant amounts of 'organic/Cl' particles with diameters smaller than 1 µm.

We applied STXM/NEXAFS to derive the individual particle MS, OVF, and speciation of the organic matter. Particle
numbers examined are significantly lower than for CCSEM/EDX analysis (Table 2). As such the corresponding uncertainties in particle MS and OVF distributions are greater. For this analysis, with a 95% confidence, the uncertainty in the representative distribution is between 20-40% (Thompson, 1987). For example, if the 30% of particles in a size bin have been identified as OC, the uncertainty is about ±4.5%. The aim of the STXM/NEXAFS analysis is not to provide the most accurate physicochemical representation of the ambient particle population but to obtain additional information of the aerosol
population serving as INP source. Figure 3 displays the size-resolved particle MS and OVF and representative NEXAFS spectra. Figure S4 displays representative false colored MS and OVF images of the particle samples for the purpose of visualization. Figure S5 presents the same data as Fig. 3, but the size-resolved particle MS and OVF are expressed as the fractional particle MS and OVF per size bin. The particle MS analysis shown in Fig. 3 indicates that all samples are dominated by particles made up by OCIN. Sample Day 1 contains the smallest amount of pure OC particles. Overall, samples Day 1 and
2 have very similar characteristics in terms of particle MS and OVF. Day 1 sample shows a slightly higher fraction of OCIN particles that contain C=C bonds (EC). For both daytime samples, the OVF is dominated by particles that contain 20-40% OC. The corresponding NEXAFS spectra point to sea salt coated with OC matter that consists of carboxylic acid components (R(C=O)OH at 288.5 eV peak) and presence of carbonate (CO$_3$ at 290.4 eV). Figure S4 shows that all particles are associated with OC matter. For most NEXAFS spectra the presence of potassium is also evident. The NEXAFS spectra of samples Day
1 and 2 are similar to previous studies examining SSA with organic coatings or aged SSA (Laskin et al., 2012a;Knopf et al., 2014;Pham et al., 2017;Tomlin et al., 2021;Ault et al., 2013).





Samples Night 1 and 2 possess some distinct features compared to the daytime samples. Night 1 samples are dominated by OCIN and OCECIN particles and contains very few IN particles and larger OC particles. The OVF analysis indicates that particles containing 40-60% OC dominate the distribution. NEXAFS spectra include signatures of organic coated SSA particles

but also spectra that show the presence of C=C bonds. Night 2 samples are dominated by OCIN and OCECIN particles, where the latter is present in the largest fraction among all particle samples considered here and limited to the 6th stage of the MOUDI. Purely OC particles are a very minor constituent of this particle population. The most prominent OVF values range between 20-60%. NEXAFS spectra mostly resemble OC associated with SSA similar to previous studies (Ault et al., 2013;Pham et al., 2017), though with lesser and more contribution of C=C bonds and hydroxyl groups. Overall, the STXM/NEXAFS results

show that most of the particles are associated with OC matter at more than 20% volume fraction and that this OC matter in most cases resembles organic components previously associated with SSA particles.

**3.2 Ice nucleation experiments**

Figure 4 shows the results of the ice nucleation experiments. Plotted data points and uncertainties reflect the average and standard deviation of conducted measurements. Except for the warmest ice formation temperatures for Day 1 and 2 samples,

all ice formation was detected at water subsaturated conditions. Only for temperatures greater than 231 K IMF was observed while sample Night 2 showed DIN for the highest temperature at 230 K. All observed ice formation events occurred at $T$ and RH conditions that would not allow for homogeneous ice nucleation. In other words, ice formation was observed for temperatures greater than the maximum for homogeneous freezing and for humidity lower than the minimum humidity required for homogeneous ice nucleation (Koop et al., 2000).

Samples Day 1, 2 display very similar conditions for DIN and IMF. In fact, they agree with each other within the measurements' uncertainties. Both samples show the greatest physicochemical similarity among the four particle samples as discussed above. Also, the particle loading is similar (Table 2). For temperatures below 240 K the Day 1, 2 samples display IMF at below 130% RH$_{ice}$, or about 85% RH, and DIN for below 121% RH$_{ice}$, or about 74% RH, both reflect subsaturated conditions. IMF at saturated conditions commenced at 240 K. Overall, these measurements indicate that the Day 1, 2 samples

could act as INPs at colder mixed-phase cloud temperatures and cirrus cloud temperatures.

Samples Nights 1, 2 show different IMF and DIN conditions. At 240 K, IMF occurs at subsaturated conditions. Sample Night 1 initiates IMF at ~231 K similar to Day 1, 2 samples at ~130% RH$_{ice}$. For lower temperatures, DIN proceeds at higher RH$_{ice}$ compared to Day 1, 2 samples. Night 2 sample induces DIN at ~231K, at ~143% RH$_{ice}$, and for all lower temperatures at greater RH$_{ice}$.

IMF and DIN proceed at similar conditions as observed in our previous studies applying particles samples collected at the Pico Mountain Observatory (PMO) in the Azores Islands (China et al., 2017;Lata et al., 2021). Since the particle samples are dominated by OCIN particles and display organic coatings (Fig. 3 and S4), we have plotted the glass transition point ($T_g$) of various organic aerosol (OA) surrogates (Charnawskas et al., 2017;Wang et al., 2012a) in Fig. 4. It is recognized that solid (glassy) particles can act as INPs (Murray et al., 2010;Wang et al., 2012a;Knopf et al., 2018). A measure of the viscosity of





the OC particle fractions is given by the $T_g$ and its full deliquescence RH (FDRH) (Knopf et al., 2018;Charnawskas et al., 2017). Lata et al. (2021) have shown the presence of highly viscous particles with $T_g$ values greater than room temperature in samples collected at PMO. The $T_g$ of laboratory generated α-pinene SOA displays the lowest $T_g$ (green line, (Charnawskas et al., 2017)), followed by naphthalene SOA (blue line, (Charnawskas et al., 2017)) and field-derived SOA $T_g$ (red line, Wang et al. (2012a)), and $T_g$ of Suwannee River Fulvic Acid (SRFA) particles (dark violet line, Wang et al. (2012a)). The dashed green

line displays the FDRH for 500 nm α-pinene SOA particles under the humidification/cooling rate of this experiment (Charnawskas et al., 2017). This indicates that the particle system between the solid and dashed green line consists of a solid organic particle surrounded by an aqueous organic solution. At the FDRH point, an aqueous solution droplet exists which continuous to take up more water if humidification or cooling commences (Berkemeier et al., 2014). However, OC matter present as a thin coating will reach its FDRH sooner, i.e., at lower temperatures and RH as discussed previously (Charnawskas

et al., 2017). Since $T_g$ of ambient SOA, naphthalene SOA, and SRFA are greater than of α-pinene SOA, we would expect that corresponding FDRHs are also greater. Considering these points, we cannot rule out that particulate OC matter served as IMF INPs, emphasizing the potential of OA to act as INPs (Knopf et al., 2021;Knopf et al., 2018). It is worthwhile to note that most DIN events occurred at conditions below the plotted $T_g$, implying that those, likely solid organic or organic-coated particles acted as INPs.

Figure 4 includes the estimated range of PCF for pores in the size range of 7.5 to 20 nm. The majority of observed DIN fall in this area. Since we do not have detailed particle morphological data, the occurrence of PCF remains speculative. The particles are likely engulfed by a glassy organic coating at those lower temperatures. If the organic coating is secondary in nature, one would expect a smoother surface due to uniform condensation of low-volatile organic gases compared to when the organic is of primary origin. However, the underlying solid particle core may modulate surface structures and it is known that

OC matter experiencing $T$ and RH cycles including freeze-drying, can change its morphology (Adler et al., 2013). More studies are needed to evaluate the possibility of the PCF mechanism being active on solid OA particles.

### 3.2 INP identification

21 individual INPs were identified and analyzed by SEM/EDX. All identified INPs were in the supermicron size (Table 3). Figure 5 shows exemplary SEM derived images of daytime and nighttime INPs and corresponding observed freezing

320   temperatures. The elemental composition of the INPs is given in Fig. 6 (marked by a star). Daytime INPs show the dendritic morphology typical for processed sea salt particles similar to previous observations (Knopf et al., 2014;Wang et al., 2015;Laskin et al., 2012b). This particle morphology is clearly different from the nighttime INPs. Figure 6 displays the average atomic percentages of each investigated sample (different from identified particle-type classes) and atomic percentages of identified INPs and exemplary EDX spectra. Additional EDX spectra of INPs are shown in Fig. S6. This data demonstrates

325   that the elemental composition of each INP (except INP Night 2 #6) reflects the typical particle composition on the sample. SEM/EDX analysis of the INPs allows for comparison with the particle types derived by the $k$-means cluster analysis. Table 3 provides the particle types that corresponds to the identified INP. Figure 6 indicates that INPs from samples Day 1, 2 and Night





1 possess very similar composition. Na and Cl dominate these particles in addition to OC with minor contribution of S and Mg. All these identified INPs fall in the particle-type class "sea salt". This corroborates previous studies indicating that SSA

particles with OC matter act as INPs (McCluskey et al., 2018a;DeMott et al., 2016;Prather et al., 2013b;Cornwell et al., 2021;Wilson et al., 2015;Knopf et al., 2011;Alpert et al., 2011b;Alpert et al., 2011a). This also indicates that with applied analytical methods we cannot distinguish the potential uniqueness of INPs from particles in the identified major particle-type classes. A similar conclusion was found in previous studies (China et al., 2017;Knopf et al., 2014;Lata et al., 2021). INPs of sample Night 2 show a different elemental composition (Fig. 6). The majority of those INPs are classified as "Sea salt

proc./dust/org" (Table 3). This may be the reason for the different particle morphology seen in Fig. 5. One exception is INP Night 2 #6 which is purely carbonaceous. Carbonaceous particles were not identified to make up a significant particle-type class, though STXM/NEXAFS analysis clearly shows that those particles are present but in small numbers (Fig. 3, S4, and S5). In comparison to the identified particle-type classes, the identified carbonaceous INP represents a "rare" INP type. All INP types contain significant amounts of OC and as such supports above discussion about a potentially glassy organic coating

acting as the INP. These findings further emphasize the need to better understand the role of OA particles and organic coatings on INPs (Knopf et al., 2021;Knopf et al., 2018).

### 3.2 Immersion freezing kinetics

We analyzed observed IMF events for their freezing efficiency using commonly applied approaches. It should be emphasized that derived freezing efficiencies represent the behavior of a diverse particle population. For an IMF temperature of 240 K, we

determined the activated INP fraction as given in Table 2. Complementary ambient particle size distribution (PSD) measurements are limited to the size range between 10 and 460 nm (Wang and Zheng, 2017). Considering this caveat, we can assume the presence of about 250 particles cm$^{-3}$ as a rough average of particle concentrations during the campaign period. This yields IMF INP number concentrations of about 10 to 20 INP L$^{-1}$ in accordance with the range of previous measurements of INP number concentrations (Knopf et al., 2021;DeMott et al., 2016;Mason et al., 2015;DeMott et al., 2010).

We further analyzed the IMF freezing data using the surface-based deterministic approach INAS and the surface and time dependent approach of CNT expressed in our ABIFM following closely our previous methodologies (China et al., 2017;Wang et al., 2012b;Wang et al., 2012a). As outlined in China et al. (2017), $J_{het} = N_{ice}/(t \cdot A_{tot})$ and $n_s = N_{ice}/A_{tot}$, where $N_{ice}$ is the number of observed ice formation events, usually equals to one; $A_{tot}$ is the total particle surface area accessible in the experiment (Table 2); $t = 12$ s is the time between two optical images during the experiment. Figure 7 shows the corresponding

$n_s$ and $J_{het}$ values as a function of the $\Delta a_w$ for the four samples as solid symbols. Note, the greater $\Delta a_w$ the closer one approaches the homogeneous freezing limit (Koop et al., 2000;Knopf and Alpert, 2013). The IMF $J_{het}$ and $n_s$ values can be readily expressed as a function of temperature and droplet $a_w$ (e.g., $a_w = 1$ for supercooled water) by application of the water-activity based ice melting curve (Koop et al., 2000;Knopf and Alpert, 2013). The uncertainty in $\Delta a_w$ accounts for the uncertainty in temperature and RH. The uncertainty in $J_{het}$ accounts for the uncertainty in surface area (Table 2) and statistical

uncertainty (Knopf et al., 2020), where the latter dominates the overall uncertainty. In addition, the right y-axis of Fig. 7





indicates the corresponding $n_s$ values. Since INAS is deterministic, thus lacks the additional parameter of time, the $n_s$ values are about a factor of 15 greater than the corresponding $J_{het}$ values. Some of the nighttime data show greater $n_s$ and $J_{het}$ values. This is, as shown in Fig. 5, because some of these samples induced IMF at subsaturated conditions at highest temperatures.

We derive an IMF parameterization for these remote marine boundary layer particle samples and for comparison with previous IMF parameterizations. Considering overall uncertainties, we merge all IMF data to derive a new $J_{het}$ and $n_s$ parameterization as a function of $\Delta a_w$. The parameters for the ACE-ENA ground site (GD), $J_{het}$ and $n_s$ fit, shown as the black solid line in Fig.7, are given in Table 4. The linear regression is performed without weighing of the data uncertainty due to the asymmetric error bars. Derived fit parameters (Table 4) reflect the approximate mean when performing the linear regression with maximum and minimum symmetric errors. The parameterization is valid for the $\Delta a_w$ range of measurements shown in

Fig. 7. Our derived $J_{het}$ and $n_s$ values are larger than the those derived from particles collected at the PMO during previous studies. However, it agrees well with very recently published IMF $J_{het}$ values for SSA derived from ambient measurements (Cornwell et al., 2021). That study derived median $J_{het}$ values of about 1000 cm$^{-2}$ s$^{-1}$ at about 243 K (with 1-2 orders of magnitude uncertainty). This corresponds to $\Delta a_w \approx 0.25$ at saturated conditions ($a_w = 1$). Our ABIFM model (Table 4) derives $J_{het}$ = 1270 cm$^{-2}$ s$^{-1}$ agreeing with their reported median $J_{het}$ values. Additionally, our IMF $J_{het}$ parameterization expands the

temperature and humidity range of applicability since Cornwell et al. (2021)'s parameterization is constrained to water saturation and temperatures from 240.15 K to 244.65 K.

    In general, IMF freezing rates from ambient particles of the Azores region display a shallower slope compared to measurements of INPs with laboratory surrogates. It can be expected that ambient aerosol contains a range of different INP types and sizes and, as such, freezing efficiencies may be broader and shallower (Knopf et al., 2020;Alpert and Knopf, 2016).

Figure 7 demonstrates that natural desert dust, employed in the freezing experiments by Niemand et al. (2012), consists of different mineral dust types. Its slope is shallower than single-component INP system studied in the lab, as exemplary shown in Fig. 7 for illite dust particles, diatom cells, and leonardite particles. Ambient particles being more diverse than desert dust particles display the shallowest slopes. This is an important feature because, consequently, these ambient particles are more efficient INPs at warmer temperatures compared to efficient single component INP-types studied in the laboratory, but less

efficient INPs at colder temperatures.

**3.3 Deposition ice nucleation kinetics**

For the analysis of DIN, we first employ CNT closely following the approach by (Wang and Knopf, 2011). Then we derive $J_{het}$ values as a function of $\Delta a_w$ allowing for an alternative parametrization following the analysis in China et al. (2017). Experimentally derived $J_{het}$ values for DIN are shown in Fig. 8a. $J_{het}$ values range between 1000 to 10000 cm$^{-2}$ s$^{-1}$. These $J_{het}$

values are about one order of magnitude greater than for IMF. Experimental $J_{het}$ values allow for the derivation of the contact angle, $\theta$. $J_{het}$ is expressed as (Pruppacher and Klett, 1997)

$$J_{het} = Ae^{\left(\frac{-\Delta F_{g,het}}{kT}\right)},$$ (1)





where $k$ is the Boltzmann constant and the free energy of ice embryo formation is defined as (Pruppacher and Klett, 1997)

$$\Delta F_{\text{g,het}} = \frac{16\pi M_w^2 \sigma_{i/v}^3}{3[RT\rho \ln S_{\text{ice}}]} f(m, x) \,, \tag{2}$$

where $R$ is the universal gas constant, $M_w$ is the molecular weight of water, $\sigma_{i/v}$ is the surface tension at the ice-vapor interface, $\rho$ is the density of ice, $S_{\text{ice}}$ is the ice saturation ratio, and $f(m, x)$ is the geometric factor. $f$ depends on $m$ and $x$. $m$ is the compatibility factor and $x$ represents the ratio of the radius of the substrate to the radius of spherical ice germ. Since the INPs are >0.1 µm, we can neglect $x$. Then, $\lim_{x \to \infty} f(m, x) = \frac{m^3 - 3m + 2}{4}$. The contact angle, $\theta$, is defined as $\cos\theta = m$ (Fletcher, 1958). The smaller the $\theta$, the more efficient the INP.

$\theta$ is plotted in Fig. 8b. Daytime samples display lower $\theta$ values compared to nighttime samples, in line with Fig. 4 showing daytime samples form ice at lower supersaturation than nighttime samples. Following Wang and Knopf (2011) we parameterize $\theta$ as a function of $RH_{\text{ice}}$ which allows to describe several DIN data sets. Figure 8c shows the contact angles for the daytime and nighttime samples as a function of $RH_{\text{ice}}$. This analysis demonstrates that daytime samples display lower $\theta$, and as such are more efficient INPs. The derived $\theta$ values closely follow the parameterization (solid black line in Fig. 8c)

given by Wang and Knopf (2011). This suggests that this parameterization can also be applied to estimate $\theta$ and thus $J_{\text{het}}$ for ACE-ENA ground site particle samples.

As an alternative parametrization we apply the water activity criterion to describe $J_{\text{het}}$ for DIN. This allows for a computationally efficient way to describe DIN. This approach was introduced in China et al. (2017) showing agreement for measurements of kaolinite INPs with previous literature data. Figure 9 presents $J_{\text{het}}$ and $n_s$ values for DIN as a function of

$\Delta a_w$. As can be seen, $J_{\text{het}}$ and $n_s$ can be well represented by a linear fit given the experimental uncertainties. As in the case for IMF, the linear regression is performed without weighing of the data uncertainty due to the asymmetric error bars. The corresponding fit parameters are given in Table 4. As evident from the CNT-based analysis shown in Fig. 8, daytime samples display the smallest contact angles indicating greatest DIN efficiency. Here, daytime samples are associated with lowest $\Delta a_w$ values. $J_{\text{het}}$ and $n_s$ values as a function of $\Delta a_w$ demonstrate an exponential behavior similar to the case of IMF (Fig. 7 and

Fig. 3 in Knopf and Alpert (2013)).

**4. Atmospheric implications**

This study shows that ambient sea salt and processed sea salt particles can affect mixed-phase and cirrus cloud formation by acting as IMF and DIN INPs. IMF and DIN conditions and corresponding nucleation rate coefficients and INAS densities are of similar order of magnitude as other investigated INP types (Knopf et al., 2018;Murray et al., 2012;Kanji et al.,

2017;Hoose and Möhler, 2012;Wang et al., 2012b;Wang and Knopf, 2011). For DIN, the contact angle for daytime samples is similarly low as for some mineral dust and clay particles (Wang et al., 2012b;Wang and Knopf, 2011). Hence, these data suggest that ambient sea salt and processed sea salt particles could serve as competitive DIN INPs under cirrus cloud conditions. Note that only a very minor fraction of particles was identified as inorganic and none of the identified INPs were



purely inorganic in nature. This corroborates previous studies that indicated the ocean can serve as a source of INPs (Creamean
et al., 2019;McCluskey et al., 2018a;McCluskey et al., 2018b;McCluskey et al., 2017;DeMott et al., 2016;Knopf et al.,
2011;Alpert et al., 2011b;Alpert et al., 2011a;Ladino et al., 2016;Wilson et al., 2015;Prather et al., 2013a;Schnell, 1975;Irish
et al., 2019;Irish et al., 2017;Knopf et al., 2014;Ickes et al., 2020;Wilbourn et al., 2020;Wolf et al., 2019;Schnell and Vali,
1975;Roy et al., 2021;Wagner et al., 2021;Cornwell et al., 2021).

We derived IMF and DIN parameterizations based on $\Delta a_{\mathrm{w}}$. $\Delta a_{\mathrm{w}}$ as a determinant of freezing, combines temperature and
RH parameters via the ice melting curve (Koop et al., 2000). As such, derived IMF and DIN parameterizations are applicable
for subsaturated and saturated conditions. Homogeneous ice nucleation is also well described by $\Delta a_{\mathrm{w}}$ (Koop et al., 2000).
Hence, applying the same $\Delta a_{\mathrm{w}}$ framework allows for a computationally efficient description of IMF, DIN, and homogeneous
freezing. This could make it a method of choice for cloud-resolving model application (Knopf et al., 2021;Fan et al.,
2017;Knopf et al., 2018). The parameterizations presented here will allow to estimate INP number concentrations for sea salt
particles dominated air masses in a remote marine region for mixed-phase and cirrus clouds conditions while accounting for
competing influence of homogeneous freezing.

## 5. Conclusions

Particle samples collected during daytime and nighttime in the remote marine boundary layer in the eastern north Atlantic were
examined for the particle population composition, INPs, and IMF and DIN. Micro-spectroscopic single-particle analyses
indicate that the sampled particle population is made up of four particle-type classes that all contain considerable amounts of
organic matter. The particle types include 'fresh sea salt particles' and 'processed sea salt particles' with varying degree of
sulfur and dust inclusions. The dominance of sea salt in observed particle types is in order considering that particle collection
was conducted in a remote marine region and close to the island's shoreline. All but one out of 21 identified INPs, do not
display different morphology or composition than the particle-type classes making up the sampled particle population. In other
words, application of nanoscale analytical tools cannot resolve the difference between the major particle-type classes and the
INPs, similar to findings of previous studies (Knopf et al., 2018;China et al., 2017;Knopf et al., 2014;Lata et al., 2021). This
is difficult to reconcile in a singular hypothesis/deterministic concept of ice nucleation but may be resolved assuming CNT.
For example, assuming the major particle-type classes possess, in average, very similar surface features. Then $J_{\mathrm{het}}$ acts on the
total particle surface area until experimental conditions (supersaturation and time) allow for the detection of ice formation,
without the need of specifying which of the particles acted as INP (Knopf et al., 2021;Knopf et al., 2020;Knopf et al., 2014).
This hypothesis will need further exploration (Knopf et al., 2021;Knopf et al., 2020).

IMF and DIN experiments point to the role of solid organic coatings as serving as INPs, though more compositional and
kinetic details about the nature of the organic coating are needed to simulate observed ice formation (Knopf et al.,
2018;Charnawskas et al., 2017;Berkemeier et al., 2014). Observed DIN proceeded in a range expected of PCF for pores in



sizes of 7.5 to 20 nm (Marcolli, 2017, 2014). Since the nanoscale morphology of the organic coating is not well-established in our experiments, it remains unclear if PCF can act as the underlying freezing mechanism.

IMF and DIN experiments yielded INP activated fraction and $J_{het}$ and $n_s$ values. We derived IMF and DIN parameterizations to express $J_{het}$ and $n_s$ for application in cloud and climate models. The derived freezing kinetics corroborate previous findings that marine derived aerosol particles can serve as INP under mixed-phase and cirrus clouds conditions.

Parameterized IMF $J_{het}$ values agree with recently published IMF $J_{het}$ values derived from field campaigns (Cornwell et al., 2021). The similarity of IMF $J_{het}$ values from SSA acting as INPs stemming from different locations may point to the similar nature of the particulate organic matter impacting the particles' propensity to act as INPs. This study demonstrates, as have others, the oceans can serve as a source of IMF and DIN INPs that should be considered when addressing cloud microphysical processes and climate.


**Data availability.** All data needed to draw the conclusions in the present study are shown in the paper and/or the Supplement.

**Supplement.** The supplement related to this article is available online at:

**Author contributions.** DAK envisioned and supervised project, performed STXM/NEXAFS experiments and analysis, conducted ice nucleation analyses, and wrote the first draft of the manuscript. JCC collected particle samples, conducted ice nucleation experiments, and assisted in STXM/NEXAFS experiments. PW, BW, AL, RCM, MKG, JYA, MF, MAM assisted in STXM/NEXAFS experiments and analyses. DPV, SC, JMT, and KAJ conducted (CC)SEM/EDX experiments and analyses. SRR conducted backward trajectory analyses. JW oversaw sampling site and particle collection. All authors discussed
interpretation of the data and contributed to the writing of the manuscript.

**Competing interests.** Some of the authors are members of the editorial board of Atmospheric Chemistry and Physics. The peer review process was guided by an independent editor. The authors have no other competing interests to declare.

**Acknowledgements.** This study was supported by the Atmospheric System Research Program and Atmospheric Radiation Measurement Program sponsored by the U.S. Department of Energy (DOE), Office of Science, Office of Biological and Environmental Research (OBER), Climate and Environmental Sciences Division (CESD). DAK acknowledges support by the U.S. DOE grants DE-SC0016370 and DE-SC0021034. RCM and AL acknowledge by the U.S. DOE grant DE-SC0018948. JW acknowledges funding support from the US DOE grant DE-SC0020259. A portion of this research was performed on a
project award (10.46936/lser.proj.2019.50738/60000088 and 10.46936/sthm.proj.2017.49857/60006200) from the Environmental Molecular Sciences Laboratory, a DOE Office of Science User Facility sponsored by the Biological and



Environmental Research program under Contract No. DE-AC05-76RL01830. The STXM/NEXAFS particle analysis was performed at beamline 5.3.2.2 at the Advanced Light Source (ALS) at Lawrence Berkeley National Laboratory. The work at the ALS was supported by the Director, Office of Science, Office of Basic Energy Sciences, of the U.S. DOE under contract
DE-AC02-05CH11231.

**Financial support**. This research has been supported by the U.S. Department of Energy (grant nos. DE-SC0016370, DE-SC0021034, DE-SC0018948, DE-SC0020259, DE-AC05-76RL01830, DE-AC02-05CH11231).

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





**Table 1: Information about collection of particle samples including sample name, sampling dates and time periods, impactor cut-off size, impactor operation cycle, and total collection time.**

| Sample ID | Dates | Local Sampling Time | Duty Cycle on/off | Cut-Off Size / µm | Collection Time |
|---|---|---|---|---|---|
| Day 1 | 6/27/17 | 11 am – 9.45 pm | 15 min / 15 min | 0.56 | 5.5 hrs |
|  | 6/28/17 | 11 am – 7 pm | 15 min / 15 min |  | 4 hrs |
|  |  |  |  |  | Total: 9.5 hrs |
| Day 2 | 7/7/17 | 11 am – 5.45 pm | 15 min / 15 min | 0.56 | 3.5 hrs |
|  | 7/8/17 | 1 pm – 5.45 pm | 15 min / 15 min |  | 2.5 hrs |
|  | 7/10/17 | 11 am – 5.45 pm | 15 min / 15 min |  | 3.5 hrs |
|  | 7/11/17 | 11 am – 5.45 pm | 15 min / 15 min |  | 3.5 hrs |
|  |  |  |  |  | Total: 13 hrs |
| Night 1 | 7/2/17 | 11 pm – 5.30 am | 30 min / 30 min | 0.56 | 3.5 hrs |
|  | 7/3/17 | 11 pm – 5.30 am | 30 min / 30 min |  | 3.5 hrs |
|  | 7/7/17 | 11 pm – 5.30 am | 30 min / 30 min |  | 3.5 hrs |
|  | 7/8/17 | 11 pm – 5.30 am | 30 min / 30 min |  | 3.5 hrs |
|  | 7/9/17 | 11 pm – 5.30 am | 30 min / 30 min |  | 3.5 hrs |
|  |  |  |  |  | Total: 17.5 hrs |
| Night 2 | 7/14/17 | 11 pm – 5.30 am | 30 min / 30 min | 0.56 | 3.5 hrs |
|  | 7/15/17 | 11 pm – 5.30 am | 30 min / 30 min |  | 3.5 hrs |
|  | 7/16/17 | 11 pm – 5.30 am | 30 min / 30 min |  | 3.5 hrs |
|  | 7/18/17 | 11 pm – 5.30 am | 30 min / 30 min |  | 3.5 hrs |
|  |  |  |  |  | Total: 14 hrs |





**Table 2: Summary of particle sample information: Sample name, number of particles examined by scanning electron microscopy (SEM), SEM sample surface area examined, SEM determined average circular equivalent diameter, number of particles examined by scanning transmission X-ray microscopy (STXM), number of particles examined for ice formation, particle surface area involved in ice formation experiments, and INP activated fraction at 240 and 221 K assuming ambient particle concentrations of 250 particles cm$^{-3}$.**

| Sample ID | Number of particles examined by CCSEM | SEM Area examined / mm² | Average circular equivalent diameter / µm | Number of particles examined by STXM | Number or particles examined for ice formation / mm² | Particle surface estimate for ice formation / cm² | INP activated fraction f at 240 K | INP activated fraction f at 221 K |
|---|---|---|---|---|---|---|---|---|
| Day 1 | 2215 | 0.36 | 0.673 | 236 | 4781±130 | $(3.4\pm0.1)\times10^{-5}$ | $6.97\times10^{-5}$ | $6.97\times10^{-5}$ |
| Day 2 | 2251 | 0.4 | 0.833 | 205 | 4073±130 | $(4.4\pm0.1)\times10^{-5}$ | $8.18\times10^{-5}$ | $4.91\times10^{-5}$ |
| Night 1 | 3733 | 0.25 | 0.349 | 263 | 11607±418 | $(2.2\pm0.1)\times10^{-5}$ | $4.31\times10^{-5}$ | $2.87\times10^{-5}$ |
| Night 2 | 2037 | 0.4 | 0.436 | 212 | 3867±157 | $(1.2\pm0.05)\times10^{-5}$ | $8.62\times10^{-5}$ | $8.62\times10^{-5}$ |





**Table 3. Information about identified ice-nucleating particles (INPs) including scanning electron microscopy (SEM) derived averaged equivalent diameter and particle-type classification.**

| INP ID | Average equivalent diameter / μm | Classification |
|---|---|---|
| Day 1 #1 | 3.13 | Sea salt |
| Day 1 #2 | 6.74 | Sea salt |
| Day 1 #3 | 6.34 | Sea salt |
| Day 1 #4 | 10.64 | Sea salt |
| Day 1 #5 | 8.14 | Sea salt |
| Day 1 #6 | 6.25 | Sea salt |
| Day 1 #7 | 5.81 | Sea salt |
| Day 2 #1 | 8.73 | Sea salt |
| Day 2 #2 | 9.17 | Sea salt |
| Night 1 #1 | 1.13 | Sea salt |
| Night 1 #2 | 0.85 | Sea salt |
| Night 1 #3 | 1.54 | Sea salt |
| Night 1 #4 | 1.42 | Sea salt |
| Night 1 #5 | 2.09 | Sea salt |
| Night 1 #6 | 6.77 | Sea salt |
| Night 2 #1 | 1.79 | Sea salt proc./dust/org |
| Night 2 #2 | 1.18 | Sea salt proc./dust/org |
| Night 2 #3 | 1.20 | Sea salt proc./dust/org |
| Night 2 #4 | 1.68 | Sea salt proc./dust/org |
| Night 2 #5 | 1.04 | Sea salt proc./dust/org |
| Night 2 #6 | 11.96 | Carbonaceous |

845

850





**Table 4. Parameters for derivation of the immersion freezing (IMF) and deposition ice nucleation (DIN) heterogeneous ice nucleation rate coefficient coefficients ($J_{het}$) and ice nucleation active sites (INAS) density ($n_s$) as a function of the water activity criterion, $\Delta a_w$, according to $\log J_{het} = c + m \cdot \Delta a_w$ and $\log n_s = c + m \cdot \Delta a_w$ are given. LCL and UCL represent lower and upper confidence levels at 95%, respectively, for the fit parameters. RMSE indicates root mean square error of the fit.**

| Parameterization | $c$ | LCL$_c$ | UCL$_c$ | $m$ | LCL$_m$ | UCL$_m$ | RMSE |
|---|---|---|---|---|---|---|---|
| IMF $J_{het}$ | 1.933 | 1.162 | 2.704 | 4.689 | 1.272 | 8.106 | 0.3557 |
| IMF $n_s$ | 3.012 | 2.241 | 3.783 | 4.689 | 1.272 | 8.106 | 0.3557 |
| DIN $J_{het}$ | 2.213 | 1.977 | 2.45 | 4.729 | 3.376 | 6.082 | 0.2122 |
| DIN $n_s$ | 3.293 | 3.056 | 3.529 | 4.729 | 3.376 | 6.082 | 0.2122 |





**Figure 1: Representative backward trajectories for examined samples and corresponding sampling periods. Color coding indicates pressure level (hPa), temperature (K), and relative humidity (%) from left to right. Black dots mark the sampling site and the trajectory locations 5 days prior of sampling. Local times are given and mark the time from which the 10-day backward trajectories are calculated.**







**Figure 2: CCSEM/EDX derived cluster analysis for identification of particle-type classes present in ACE-ENA particle samples. Upper panel represents the cumulative atomic fraction of all analyzed particles. Lower panels give detailed views of the cluster-averaged elemental compositions of the four particle-type classes identified by *k*-means cluster analysis. The representative particle types include: 'processed sea salt with mineral dust, sulfur, and organic matter', 'sea salt particles', 'processed sea salt with mineral dust', and 'organic matter-chlorine' containing particles.**







**Figure 3: Size-resolved particle composition for particle samples determined by micro-spectroscopic single-particle analysis as a function of area equivalent diameter (AED). From left to right: first column: CCSEM/EDX particle-type classes. Second column: STXM/NEXAFS derived groups of particles with different mixing states: IN - inorganic, EC - elemental carbon, OC - organic carbon. Third column: organic volume fraction (OVF). Fourth column: representative NEXAFS spectra at the carbon K-edge.**



**Figure 4:** IMF - Immersion freezing (open symbols) and DIN - deposition ice nucleation (solid symbols) for examined ACE-ENA particle samples are given as function of $T$ and $RH_{ice}$. The data points and error bars reflect the mean of several measurements accounting for uncertainties in RH. Solid line represents conditions of water saturation (100% RH). Dotted lines indicate constant relative humidity (RH). Dashed line and grey shading represent the homogeneous freezing limit for droplets of 10 μm in size and corresponding uncertainty (Koop, 2004;Koop et al., 2000). The glass transition temperature of laboratory generated α-pinene SOA (green line or 1, (Charnawskas et al., 2017)), naphthalene SOA (blue line or 2, (Charnawskas et al., 2017)) field-derived SOA (red line or 3, (Wang et al., 2012a)), and Suwannee River Fulvic Acid (SRFA) particles (dark violet line or 4, (Wang et al., 2012a)) are plotted. The dashed green line (or 5) displays the FDRH for α-pinene SOA particles, 500 nm in diameter, under the humidification rate of this experiment (Charnawskas et al., 2017). The light bluish area indicates the conditions for pore condensation freezing for pore sizes of 7.5 to 20 nm (Marcolli, 2020, 2014).





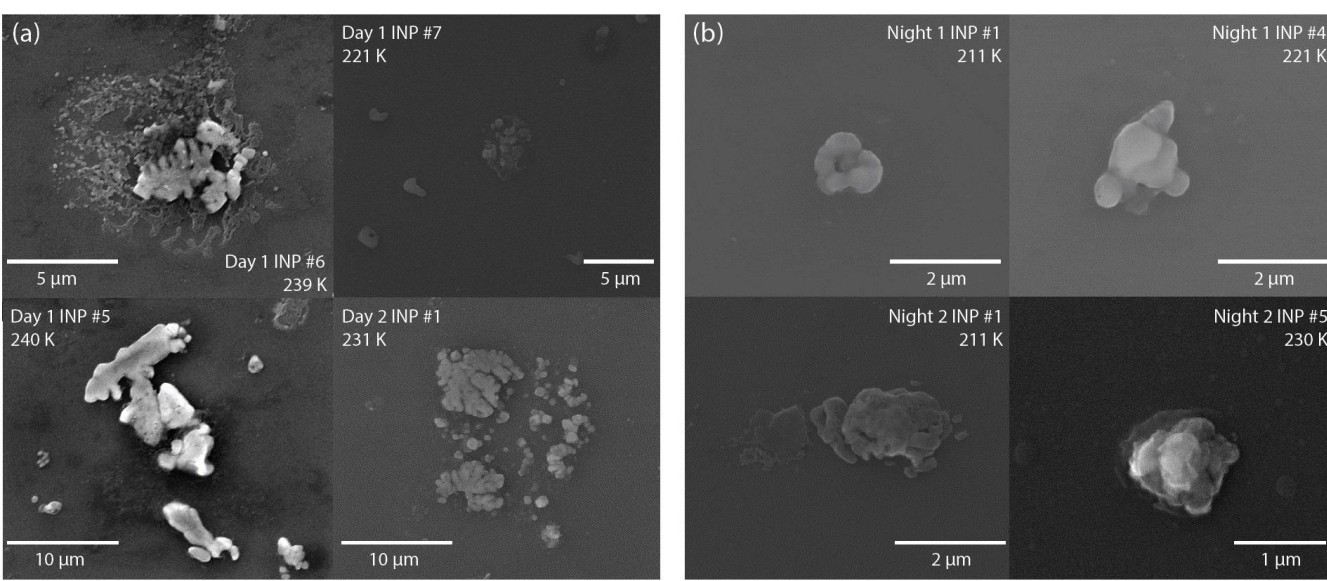

**Figure 5: Exemplary electron micrographs of identified ice-nucleating particles (INPs) for daytime particle samples (a) and nighttime particle samples (b). INP # numbers refer to INPs shown in Fig. 6. The observed ice formation temperature is given for each INP.**





**Figure 6: Composition of experimentally identified ice-nucleating particles (INPs). (a) Cumulative atomic percent of elements for 21 identified individual INPs are shown as bars for examined ACE-ENA particle samples. The first column represents the average cumulative atomic percent of elements for the specific particle sample. Stars indicate INPs shown in Fig. 5. (b) shows representative EDX spectra of identified INPs. \*corresponds to the signal from the substrate (Si₃N₄ coated silicon wafer chips) and the chamber/holder (Al).**





**Figure 7: Immersion freezing (IMF) data of examined ACE-ENA particle samples (solid symbols) and of previous studies (colored lines) as given in legend. Heterogeneous ice nucleation rate coefficients ($J_{het}$) and ice nucleation active sites (INAS) density ($n_s$) are presented as a function of the water activity criterion $\Delta a_w$. Error bars include uncertainties in temperature, humidity, and surface area. Solid and dotted black lines represent a linear regression fit and associated fit uncertainties 'ACE-ENA GD' to the data. Purple and magenta solid lines indicating 'PMO FT 2017' and 'PMO FT 2021', respectively, represent IMF parameterizations of particles collected at the Pico Mountain Observatory (PMO) under free tropospheric (FT) conditions, in the Azores on a neighboring island (China et al., 2017;Lata et al., 2021). Please note that only $J_{het}$ was reported for PMO FT 2021 (Lata et al., 2021).**





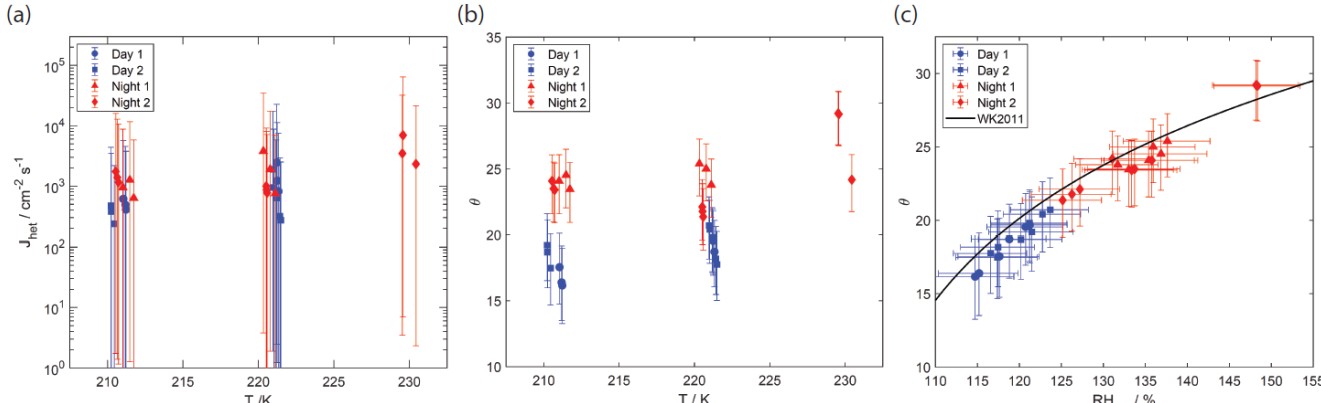

**Figure 8: Deposition ice nucleation (DIN) data of examined ACE-ENA particle samples (solid symbols). (a) Heterogeneous ice nucleation rate coefficients ($J_{het}$) as a function of temperature. (b) Contact angles ($\theta$) corresponding to $J_{het}$ values shown in (a). (c) $\theta$ values for relative humidity with respect to ice (RH$_{ice}$) under which DIN was observed. Solid line represents the DIN**
**parameterization by Wang and Knopf (2011).**





**Figure 9: Kinetic deposition ice nucleation (DIN) data of examined ACE-ENA particle samples (solid symbols). Heterogeneous ice nucleation rate coefficients ($J_{het}$) and ice nucleation active sites (INAS) density ($n_s$) are presented. Error bars included uncertainties in temperature, humidity, and surface area. Solid and dotted black lines represent a linear regression fit and associated fit uncertainties.**