# Peer review of "Micro-spectroscopic and freezing characterization of ice-nucleating particles collected in the marine boundary layer in the Eastern North Atlantic"

_Atmospheric Chemistry and Physics, 2022_

## Author Response (AR1)

**Response to Referees**

We thank both reviewers for taking the time to evaluate our manuscript. Their comments have improved this manuscript. Below we have included a point-by-point response where our comments are given in bold fonts. The line numbers refer to the revised, marked-up manuscript attached at the end. Changes in the manuscript are highlighted in red fonts.

As we worked on the response to the referees, we realized that the UTC time conversion for the backward trajectories was done erroneously by one hour. The time labeling has been corrected in revised Figs. 1 and S2. This did not change the backward trajectory results or its conclusions.

In Table 1 we omit abbreviation "f" for fraction since we do not use it in the manuscript.

**Response to Referee #1**

This manuscript is well written. The experimental study of INPs from Graciosa is rare and invaluable. Though the data is limited to a single summer season based on a few samples, the result is well presented and worth it for the science community. The authors clearly address the necessity of future study, and this reviewer agrees with the addressed outlook. The study topic is relevant to the journal scope as ACP supports many INP-related papers. This reviewer supports the publication of this paper in ACP and has only several technical comments.

**We thank the reviewer for taking the time to evaluate our manuscript and this general positive note.**

P2L53-54: Later in this manuscript, the authors mention PCF and potential pre-activation. Then, in general, the ice formation pathway can be modulated by not only ambient conditions but also particle physical properties, correct? This point can be clarified here.

**Considering the microscopic scale of ice nucleation, the reviewer makes a valid point. We will add this information and change the sentence accordingly:**

**Lines 53-55: "Furthermore, different ice formation pathways exist, which in turn depend on particle properties and ambient conditions such as temperature (T) and relative humidity (RH) (Marcolli, 2014;Pruppacher and Klett, 1997;Vali et al., 2015;Knopf et al., 2018)."**

P3L75-77: The reviewer believes that the majority of past studies focus on IMF because IMF is the dominant ice nucleation path in the atmosphere. The authors might want to briefly explain how important (dominant) DIN is compared to IMF for the reader.

**The reviewer is correct here. IMF represents the dominant ice nucleation path in the atmosphere and thus is mostly studied. DIN can become more important at lower temperatures and at water subsaturated conditions typical of upper tropospheric conditions. We add the following information to this section:**

**Lines 79-83: "IMF is recognized as the dominant primary ice formation pathway in mixed-phase cloud regimes (Ansmann et al., 2009;de Boer et al., 2011;Westbrook and Illingworth, 2013) where supercooled droplets and ice crystals can coexist. DIN can contribute to ice crystal formation at lower temperatures and water subsaturated conditions representing cirrus cloud regimes, typical of the upper troposphere (DeMott, 2002;Heymsfield et al., 2017;Cziczo et al., 2013)."**

P4L114: How low were ambient particle numbers? Does the ENA site offer the total aerosol particle concentration data?

**When writing the manuscript we missed available total aerosol concentration data at the ENA site for particles larger than 10 nm reported by (Gallo et al., 2020). We now use hourly reported aerosol concentration over the collection periods of examined particle samples. This resulted in aerosol concentrations of 477, 493, 537, and 333 cm-3 for samples Day 1, Day2, Night 1, and Night 2, respectively. This is slightly greater than originally estimated 250 cm-3.**

**We change the original sentence on line 113-115**

**"Since ambient particle numbers were low and sampling intervals for this ice nucleation study were limited, particle collection was conducted over several days intermittently."**

 **to**

**Lines 117-119: "Since ambient particle numbers were low (between 330 and 540 cm$^{-3}$, Gallo et al. (2020)) and sampling intervals for this ice nucleation study were limited, particle collection was conducted over several days intermittently."**

P4L119: Why this particular stage (D50 = 0.56 micron) was selected for sampling and subsequent analysis?

**This stage has been chosen based on our previous experience to satisfy the necessary particle loading among the different applied analytical techniques. If the particle loading is too low, ice formation might not be detectable and CCSEM/EDX and STXM/NEXAFS analysis will not result in statistically significant data sets. If the particle loading is too high, one may lose the single particle characteristics. Hence, particle samples are first interrogated and then decided which cut-off diameter is chosen that can be used across different sampling periods and analytical techniques to allow for more consistent analyses.**

**We will add on line 124 this additional information:**

**"The particle samples for a given MOUDI stage were chosen in such a way to provide the optimal particle loading for ice nucleation experiments and application of single-particle analytical techniques across the different sampling periods to allow for more consistent analyses."**

P4L129: Has the impact of precipitation been considered in the 10-day back trajectory? Heavy precipitation may have washed out aerosol particles in the given air mass (if they traveled near the surface)?

**No, precipitation has not been considered as a potential loss mechanism of the boundary layer aerosol. In our case, though, long range transport from the free troposphere largely involves descending airmasses (Fig. S2) such that precipitation is not expected along them. Also, for a well-mixed boundary layer, one could expect that aerosol, e.g., stemming from the ocean emitted by bubble-bursting processes, are dispersed throughout the boundary layer within hours to a day. This would result in a replenishment of particles. As can be seen from Figs. 1 and S2, prior to arriving at the sampling site the airmasses often remained in the boundary layer for several days.**

P5L151: It may be worth providing a reference (or brief description) of the k-means cluster method here for the reader who is not familiar with CCSEM/EDX.

**We added as a reference our previous studies and two textbooks on this matter:**

**Lines 162-164: "More than 2000 particles were examined for each sample, though for identification of the major particle-type classes via k-means cluster analysis, the CCSEM/EDX data for all samples were taken together(Moffet et al., 2013;Seber, 1984;Spath, 1985;Tomlin et al., 2021;Tomlin et al., 2020)."**

P6L187: the particle temperature - presuming the measured substrate surface temperature is equivalent to it?

**Yes, this is the case. Melting points of thin films of organics and ice display a difference to the substrate temperature of less than 0.1 K (Knopf and Rigg, 2011). For applied ambient particles (~1 μm in size, smaller than the thin films), particle temperature can be assumed to be equal to substrate temperature. However, at this point of the text it is appropriate to use "substrate temperature", thus we change sentence to**

**Lines 202-203: "RH in the ice nucleation cell is derived by the substrate temperature and the measured dewpoint (Wang and Knopf, 2011)."**

**And add on lines 205-207:**

**"The temperature accuracy of the cooling stage is independently verified by measuring the melting points of different organic compounds and ice indicating less than 0.1 K difference between substrate and particle temperature (Knopf and Rigg, 2011)."**

P10L318: All identified INPs were in supermicron size because of the image resolution limit of the optical microscope for ice nucleation experiments, or is it the nature of INPs for the samples used in this study? The reviewer is aware that the authors cite some papers (e.g., Knopf et al., 2014). Regardless, this point can be perhaps briefly clarified in the manuscript.

Indeed, the INPs were all in the supermicrometer size regime. The INPs are examined using SEM applying re-localization of the ice crystal coordinates from the optical microscope. Line 318 provides this information:

"21 individual INPs were identified and analyzed by SEM/EDX. All identified INPs were in the supermicron size (Table 3)."

To make this point clearer we change the first sentence to

Line 334: "21 individual INPs were identified and analyzed by SEM/EDX allowing for greater resolution compared to the OM.".

As other studies have observed, as discussed in our review (Knopf et al., 2018), larger ambient particles can serve as a significant source of INPs. The fact that all INPs are in the supermicrometer sized will be added to the "Atmospheric implications" and "Conclusions" sections:

Lines 449-452: "Furthermore, all identified INPs belong to the supermicrometer size regime. INP measurements at a coastal marine boundary layer site also indicated that the greatest INP number concentrations are associated with the largest particles (Mason et al., 2015). These results further emphasize the need to examine supermicrometer-sized particles for their ability to serve as INPs."

Line 467-468: "All identified INPs belong to the supermicrometer size regime."

P11L346-347: How did the authors estimate ~250 cm^-3 of aerosol particle concentration? Taking particle density in the examined cross-section of the substrate and sampled air volume to estimate it in the unit volume of sampled air? This procedure can be clarified in the text. Did the estimate show any variation in different sampling periods (i.e., D1, D2, N1, and N2)?

As outlined above we have corrected this value using complementary on-site total particle concentration measurements for particles larger than 10 nm (Gallo et al., 2020). Since we have stated the range of particle number concentrations at an earlier place in the manuscript, we change the original sentences

"Complementary ambient particle size distribution (PSD) measurements are limited to the size range between 10 and 460 nm (Wang and Zheng, 2017). Considering this caveat, we can assume the presence of about 250 particles cm-3 as a rough average of particle concentrations during the campaign period. This yields IMF INP number concentrations of about 10 to 20 INP L$^{-1}$ in accordance with the range of previous measurements of INP number concentrations (Knopf et al., 2021;DeMott et al., 2016;Mason et al., 2015;DeMott et al., 2010)."

To

Lines 361-364: "Applying mean total ambient particle concentrations for given sampling periods (Gallo et al., 2020) yields IMF INP number concentrations of about 15 to 40 INP L$^{-1}$ in accordance with the range of previous measurements of INP number concentrations (Knopf et al., 2021;DeMott et al., 2016;Mason et al., 2015;DeMott et al., 2010)."

In Table 2, in the caption, we delete the superfluous statement

**"at 240 and 221 K assuming ambient particle concentrations of 250 particles cm-3."**

Table 2: >10,000 particles per mm^2 cross-section seems plenty. Were there any particles agglomerated upon impaction on substrates and miscounted as supermicron particles?

**This is not an uncommon value for particle loading for this particle size range. Previously, we were able to work with particle loadings up to 1E6 particles per mm$^2$ (China et al., 2017;Knopf et al., 2014;Wang et al., 2012). One cannot examine on a nanometer scale the entire substrate involved in the ice nucleation experiment by SEM. This would take too much time. However, randomly chosen examination areas (smaller than 1 mm$^2$ in size) did not show particle agglomeration by double impaction. Hence, even if agglomeration occurs, one can assume it not to be significant.**

Figure S1. The ENA site is located right next to the airport and access road. Thus, there must have been some inclusions of particles from these sources on the authors' samples. This point can be addressed in the supplement figure caption.

**The location of the ENA site could be impacted by the airport and access road. However, the duration of local pollution from airport and access road is relatively short. In addition, these fresh emissions are dominated by nucleation/Aitken mode particles. Furthermore, these events are short compared to the multi-hours and multi-day sampling for presented particle samples. Hence, the airport and access road are not expected to influence the examined particle population and INPs significantly.**

**We add to the caption of Fig. S1:**

**"Airport and access road could impact particle collection. However, the duration of local pollution from airport and access road is relatively short. In addition, these fresh emissions are dominated by nucleation/Aitken mode particles outside of the examined particle and INP sizes."**

**Response to Referee #2**

I support publication. The manuscript is well written and the data is well presented.

**We thank the reviewer for taking the time to evaluate our manuscript and this positive note.**

There is one point which I think should be addressed, but overall, the manuscript more than meets the standards for publication in ACP.

The surface area of the particles that are examined in this study is a critical part of the analysis, and there are some subtleties to this that need to be explained more carefully.

If I am reading the manuscript correctly, all of the particles are from stage 6 of a MOUDI, which has a cut size of 0.56 micrometers. I realize of course that the cut size isn't a step function. But that cut size and the size of the particles listed in Table 3 are not consistent. For example, the last particle listed (Night 2 #6) has an area equivalent diameter of 11.96 microns. The MOUDI's largest cut size is 10 microns, if I'm using the sizes for the right model here. I could certainly understand a particle that large being on the first stage. But the sixth? This is only the most egregious example. Most of the other particles have sizes larger than the cut size that's listed in the manuscript.

**As evident in our numerous previous publications using MOUDI, the theoretical cut-off size does not limit the size range of collected aerosol particles in an exact manner. The reasons for this are due to particle bounce and non-sphericity and varying density of the particles. To inhibit particle bounce it is typically suggested a silicon oil spray is applied on the substrates (Marple et al., 1991). Obviously, this would contaminate our particle samples. The cut-off sizes are derived for spherical particles with a specific aerodynamic particle diameter (and particle density). The collected (ambient) particles are clearly not perfectly spherical and uniform in density. This will further shift the cut-off curve of a given stage. Lastly, the sigmoidal cut-off curve for stage 6 extends the 100% collection efficiency close to 1 μm particle diameter (Marple et al., 1991). Thus, small variations in the aerodynamic diameter will shift the cut-off size significantly to larger particles.**

**Other effects that could change the particle sizes on a substrate include particle shattering that can occur for marine derived particles (Mouri and Okada, 1993;Pham et al., 2017) which would lead to the presence of smaller particles. If particles impact as liquid droplets, they might spread out. This may be observed as a halo surrounding the particle. We did not encounter such instances. Lastly if the sample is overloaded, particle may overlap. This was also not the case of the samples examined in this study.**

**In this sampling location, collection occurred over several hours and days. Hence there is the chance to encounter particle sizes larger than given by cut-off diameter due to particle bounce and non-sphericity of particles. For this reason, we always perform SEM analyses to derive particle sizes for a given particle sample and do not rely on the cut-off size of the specific stage.**

**We will add the following information to section "2.1 Particle Sampling":**

**Lines 126-128: "Particle bounce, shattering, and non-sphericity of the ambient particles can lead to a much wider range in particle sizes collected on a stage with specific cut-off diameter (Marple et al.,**

**1991;Knopf et al., 2014;Pham et al., 2017;Mouri and Okada, 1993). Hence, single-particle micro-spectroscopic analyses are used to determine the geometric particle sizes on the sample substrate."**

This discrepancy or uncertainty can also be seen in the figures, for example Fig. 3. The size distributions are shown as a function of area equivalent diameter. In that figure, the axes extend only to 2 microns or so, but the distributions are all showing particles larger than the MOUDI cut size. (See for example, Day 1, STXM mixing state… the mode of the distribution is at about 1 micron.) These discrepancies are not as large, but they are still puzzling.

**Please see our previous response. This is due to the combined effects of cut-off curve efficiency, particle bounce, shattering, and non-sphericity.**

I realize that the MOUDI cut size and the area equivalent diameter derived from SEM measurements are not the same thing. But the authors do make an explicit point that these particles come from stage 6 of the MOUDI, and the cut size is specified. If you then go on to discuss particles much larger than the cut size, at least comment on it and perhaps provide some rationale for it.

**We emphasize that particles examined in this study are from one stage only. For a complete description of the entire aerosol population, one would need to examine more particle samples acquired from different stages. Hence, the emphasis is on the application of a single stage in this study. We are working on a manuscript that examines particles collected on substrates from different stages. Here, we analyze the particles on the substrate that were also used for the ice nucleation experiments. Thus, we do not claim that the particle population examined reflects the entire ambient particle population. Please see also our response to the first comment.**

The larger issue is the derivation of the surface area of the particles. (This issue is highlighted repeatedly in Knopf et al, 2020, which is cited in the manuscript.) Deriving the surface area of the particles from the SEM images (the area equivalent diameter) is problematic. Is the assumption that the particles are spherical, then using that to get a surface area once a diameter is derived? This is never stated.

**This is a very good point raised by the reviewer. We agree to clearly communicate the assumptions underlying the particle surface area employed in the analysis of the data.**

**Indeed, despite the use of SEM, the surface area available for ice nucleation given in Table 2 can be considered to be an estimate. However, it is still the most feasible approach considering the large numbers of different particle types and morphologies that are probed during the analyses. The equivalent area diameter represents the diameter of a circle that would equal the same projected surface area as the imaged particle. We then assume (half-) spherical particles to estimate the total particle surface area available in the ice nucleation experiments. On a nanoscale, this clearly will not reflect the actual particle surface area.**

**We will add this information on lines 160-161:**

"The equivalent circle diameter represents the diameter of a circle that would equal the same projected surface area as the imaged particle."

On lines 169-171, we add:

"Estimation of particle surface area applies an equivalent circle diameter and relies on the assumption that the particles are spherical. The collected particles are in most cases not spherical. Furthermore, nanoscale morphology, like nanopores, cavities, and cracks, is not considered in this analysis."

And if that is, in fact, what is assumed, it almost certainly wrong. The SEM only "sees" the top of the particle. You can't access the third dimension. There's been work in recent years showing that dust, for example, is rarely spherical. See Huang et al, 2020 for example. I know that the INPs detailed here are not dust, but the point is still valid. None of the particles in panel A of Fig. 5, for example, look spherical to me.

**The reviewer is correct in this point. When analyzing 1000s of particles by CCSEM/EDX and avoiding beam damage (including possible volatilization of organic matter), one cannot apply higher resolution and/or shift the sampling plane to assess the third dimension.**

**The implications for the freezing kinetics could be twofold: If we overestimate the particle surface area by assuming particle sphericity, then derived $J_{het}$ values would represent lower limits and ice formation in the atmosphere may proceed more efficiently. Considering the presence of nanopores, cracks and cavities on the order of nanometers, we likely underestimate the actual particle surface area. In this case, our derived $J_{het}$ values would represent upper limits. However, one should keep in mind that the statistical uncertainty of nucleation is about +1 order of magnitude and about -3 orders of magnitude. This uncertainty range might cover the uncertainty in surface area by neglecting nanoscale features and three dimensionality.**

**We will add this discussion on lines 375-380:**

**"As outlined above, particle surface area is derived assuming that the particles are spherical and neglecting nanoscale morphological features. If the particle surface area is overestimated, derived $J_{het}$ values would represent lower limits. Taking into account nanoscale morphology, which was not achievable in this study, might yield greater particle surface area. This in turn, would indicate the underestimation of the actual particle surface area. Hence, derived $J_{het}$ values would represent upper limits. However, the statistical uncertainty of nucleation spans several orders of magnitude and, thus, likely accounts for the uncertainty in particle surface area."**

I am not asking the authors to resolve these issues. I am asking that they more explicitly outline how the particle surface area is derived and discuss (at least) some of the uncertainties that may arise from that method. (I think quantifying the uncertainty may be beyond what's possible for this study.)

**We hope our responses above are satisfactory. We now mention the assumption of particle sphericity and the potential uncertainties related to this. Considering the nature of the particles containing organic matter, which is more prone to beam damage, and the number of particles examined, the**

**reviewer is correct that a high-resolution determination of the particle surface area is not possible in this study. Understanding the nature of the INPs in relation to the particle population was the primary goal of this study. Decreasing the uncertainty in surface area significantly for these large particle samples would require a significantly greater effort which is unlikely to change the conclusions made.**

Minor points

Table 2, heading to column 6: ''or'' should ''of''?

**Yes, a typo. It should be "of". This is corrected.**

Figure 4: the blue open circle at ~ 231 K... The open symbol indicates that this is immersion freezing, but it is well below the water saturation line. Is this a particle that deliquesced, then froze from solution?

**This is a correct interpretation. Previous work by Molina, Koop, our group and others have shown that immersion freezing can proceed in subsaturated (with respect to water) conditions. One can envision that a hygroscopic coating was present (e.g., NaCl) that could result in an aqueous solution surrounding the INP from which ice formed (e.g., deliquescence of NaCl at low temperatures, (Wagner et al., 2018)). Another scenario could include a partially deliquesced amorphous solid organic particle (or solid organic coating) that served as INP. See, e.g., (Berkemeier et al., 2014;Knopf et al., 2018).**